

# Delivery by caesarean section and risk of childhood obesity: analysis of a Peruvian prospective cohort

Rodrigo M. Carrillo-Larco[1,2], J. Jaime Miranda[1,2] and Antonio Bernabé-Ortiz[1,3]

[1] CRONICAS Center of Excellence in Chronic Diseases, Universidad Peruana Cayetano Heredia, Lima, Peru
[2] School of Medicine, Department of Medicine, Universidad Peruana Cayetano Heredia, Lima, Peru
[3] Escuela de Medicina, Universidad Peruana de Ciencias Aplicadas, Lima, Peru

## ABSTRACT

**Objectives.** We aimed to assess if Caesarean section is a risk factor for overnutrition in early- and late-childhood, and to assess the magnitude of the effect of child- versus family-related variables in these risk estimates.

**Methods**. Longitudinal data from Peruvian children from the Young Lives Study was used. Outcomes assessed were overweight, obesity, overnutrition (overweight plus obesity), and central obesity (waist circumference) at the age 5 (first follow-up) and 7 (second follow-up) years. The exposure of interests was delivery by Caesarean section. Relative risks (RR) and 95% confidence intervals (95% CI) were calculated using multivariable models adjusted for child-related (e.g., birth weight) and family-related (e.g., maternal nutritional status) variables.

**Results.** At baseline, mean age was 11.7 ($\pm$ 3.5) months and 50.1% were boys. Children born by Caesarean section were 15.6%. The 10.5% of the children were overweight and 2.4% were obese. For the obesity outcome, data from 6,038 and 9,625 children-years was included from baseline to the first and second follow-up, respectively. Compared to those who did not experience Caesarean delivery, the risk of having obesity was higher in the group born by Caesarean: RRs were higher at early-childhood (first follow-up: 2.25; 95% CI [1.36–3.74]) than later in life (second follow-up: 1.57; 95% CI [1.02–2.41]). Family-related variables had a greater effect in attenuating the risk estimates for obesity at the first, than at the second follow-up.

**Conclusion.** Our results suggest a higher probability of developing obesity, but not overweight, among children born by Caesarean section delivery. The magnitude of risk estimates decreased over time, and family-related variables had a stronger effect on the risk estimates at early-childhood.

## INTRODUCTION

Childhood obesity is a global public health issue, and characterization of its early determinants is necessary to tackle this problem (*Rooney, Mathiason & Schauberger, 2011*).

Corresponding author
Rodrigo M. Carrillo-Larco, rodrigo.carrillo@upch.pe

Delivery mode has gained attention as Caesarean section has been reported as a risk factor for childhood obesity in some (*Goldani et al., 2011*; *Goldani et al., 2013*; *Huh et al., 2012*; *Keag, Stock & Norman, 2014*; *Mueller et al., 2015*; *Pei et al., 2014*; *Rooney, Mathiason & Schauberger, 2011*; *Zhou et al., 2011*), but not all studies (*Ajslev et al., 2011*; *Barros et al., 2012*; *Darmasseelane et al., 2014*; *Flemming et al., 2013*; *Goldani et al., 2013*; *Kuhle, Tong & Woolcott, 2015*). These dissimilar results could be explained by different characteristics of the study population, different patterns and distribution of the exposure or outcome in the study setting, and also due to the age at which the outcome was assessed.

Although a recent meta-analysis suggested there was no difference in the pooled relative risk between studies assessing the outcome when the child was younger than 6 years versus at age 6 or older, other report concluded that the magnitude of the association between Caesarean section and obesity appears to be stronger at younger ages (*Pei et al., 2014*). Furthermore, other meta-analysis assessing the outcome at adulthood reported weaker association estimate in comparison to that one reported for children aged 6 or older (*Darmasseelane et al., 2014*; *Kuhle, Tong & Woolcott, 2015*). Overall, the association between Caesarean section and obesity seems to become weaker as the subject becomes older. *Pei et al. (2014)* suggested this could be because other risk factors become more important for developing obesity, as the child grows up: diet and physical activity patterns, as well as the surrounding environment; overall, these are child-related variables. Consequently, we hypothesize that other variables not directly related to the children, such as maternal education or nutritional status, play a more important role during early childhood, and this effect would diminish over child-related variables as the child grows older. A hierarchical approach including child- and family-related variables separately would help to have further evidence on this hypothesis.

Although there is a growing body of literature regarding the association of Caesarean section and childhood obesity, some challenges remain (*Weng et al., 2012*). For example, further prospective studies should verify if there is actually no difference in the risk estimates assessed at age 6, younger or older. Prospective studies from different world regions are needed (*Batty, Victora & Lawlor, 2009*), as developing-country settings may have a different distribution of exposures (*Ebrahim et al., 2013*) that may influence the strength of association between Caesarean section and childhood obesity, e.g., diet patterns, infections, and antibiotics use. To date, none of the available evidence on the matter does come from Latin American settings (*Kuhle, Tong & Woolcott, 2015*). Furthermore, most of the studies assess obesity or overweight as per body mass index (BMI) (*Kuhle, Tong & Woolcott, 2015*), thus providing room to complement current knowledge with central obesity, as per waist circumference, as the outcome of interest. Therefore, studying the association between Caesarean section and offspring obesity becomes relevant, especially in settings where the number of Caesarean sections performed has increased in recent years (*Quispe et al., 2010*).

The aim of this study was to assess if Caesarean section is a risk factor for overnutrition in early- and late-childhood, and to assess the magnitude of effect of child- versus family-related variables in these risk estimates. For this we will follow a hierarchical

approach including child- and family-related variables to assess whether these variables does alter or not the risk estimates at two points in time, early- and late-childhood.

## METHODS

### Study design and setting

Secondary data analysis from the Young Lives study (*Barnett et al., 2013*), an international prospective cohort study conducted in four developing countries: Ethiopia, India, Peru and Vietnam. The Young Lives study started in the year 2002 with two prospective cohorts. At baseline, the younger cohort included children aged 6–18 months; whilst the older cohort recruited children aged 7–8 years.

For this analysis we used data of the younger Peruvian cohort; we used data of the first (hereafter known as *baseline*), second (hereafter known as *first follow-up*), and third (hereafter known as *second follow-up*) round of the younger Peruvian cohort. These rounds were conducted in the years 2002 (children mean age: $11.7 \pm 3.5$ months), 2006–07 (children mean age: $5.3 \pm 0.4$ years) and 2009–10 (children mean age: $7.9 \pm 0.3$ years), when participants were at infancy, early- and late-childhood, respectively (*Barnett et al., 2013*).

We did not include the other countries because preliminary analyses revealed a small number of cases of obesity during follow-up, thus preventing us from conducting multivariable analysis. In addition, we only used the younger Peruvian cohort because important variables, such as prematurity, were not available in the older Peruvian cohort.

### Participants

For the younger Peruvian cohort, the attrition rate was 4.3% and 1.1% at the first and second follow-up, respectively; further details about the attrition rates have been published elsewhere (*Barnett et al., 2013*).

For this secondary analysis, children with history of prematurity were excluded. Participants included in the analysis were those with complete information in the variables of interest: child BMI, age, sex, and delivery mode. The original study enrolled 2,052 children, 549 were excluded because of prematurity and 529 due to incomplete data. Thus, 974 children were included in the analysis at baseline (Fig. S1).

Previous studies set singleton pregnancy as an inclusion criterion (*Goldani et al., 2013*; *Huh et al., 2012*). Since multiple gestation can lead to preterm delivery (*Goldenberg et al., 2008*; *Slattery & Morrison, 2002*), we decided to exclude premature children. Previous studies also excluded premature children, so we decided for this exclusion criteria in order to ensure comparability (*Huh et al., 2012*; *Mueller et al., 2015*; *Pei et al., 2014*).

### Variables

Primary and secondary outcomes, as well as exposure and other variables, and when these were assessed, are presented in Fig S2. All the data was collected from participants with standardized questionnaires, and weight, height, as well as waist circumference were obtained with adequate anthropometric techniques (*Barnett et al., 2013*).

### Outcomes

The primary outcomes were overweight and obesity. Additionally, we explored two secondary outcomes: overnutrition (overweight plus obesity) and central obesity. All these variables were assessed at the first and second follow-up. However, central obesity was only assessed at the second follow-up.

Overweight and obesity were defined as per BMI. Two different criteria to set BMI into categories were used. For children at baseline we defined overweight and obesity using $z$-score of weight-for-length: more than two but less than three $z$-score was overweight, and three or greater $z$-score was obesity. Regarding children at the first and second follow-up, overweight and obesity were defined with age- and sex-specific cut-off points proposed by the International Obesity Task Force (IOTF) (*Cole & Lobstein, 2012*). We used the IOTF criteria because it gives more conservative rates of obesity in comparison to other international definitions (*Gonzalez-Casanova et al., 2013*; *Padula & Salceda, 2008*; *Ramirez et al., 2006*). However, the IOTF has not set cut-off points for children younger than two years old. This forced us to use $z$-score of weight-for-length for younger children so to characterize them as either overweight or obese and exclude them for the incidence and risk analysis.

Central obesity was defined using waist circumference based on the cut-offs proposed by *Bergmann et al. (2010)*. These cut-off points were chosen because they increased the risk for cardiovascular risk factors among children: 63.85 cm for boys younger than 8 years old and 64.00 cm for those with eight years or older; for girls: 58.25 and 59.65 cm, respectively.

### Exposure

The exposure of interest was delivery mode collected at baseline, yet mothers had to recall this information as no birth certificate was requested. This information was retrieved with the question: *was the child delivered by a Caesarean section?* (Yes/no).

### Other variables at baseline

Co-variables were assessed at baseline and included information on both the children and mothers. In the case of children, variables included: sex (boy or girl), age at baseline (<1 years or ≥1 year), birth weight (2,500 g–4,000 g, ≤2,500 g, ≥4,000 g), hospitalization immediately after birth (*Was the child hospitalized after the birth?*, Yes/no), birth order (1st, 2nd or ≥3rd), and breastfeeding (*Did you ever breastfed the child?*, Yes/no); although breastfeeding could be in the causal path from Caesarean section and childhood obesity, the effect of such variable needs to be controlled for in the risk estimates; therefore, we included it in the regression models (see statistical analysis section). Other co-variables included in the analyses were: maternal nutritional status by BMI at baseline (normal weight (≥18.5 and <25), overweight (≥25 and <30), and obesity (≥30)); maternal educational attainment (none/primary, high school, higher education); household wealth index (in tertiles); and household location (rural or urban).

## Sampling and procedures

The sampling design as well as the collection methods are available online. Briefly, the Peruvian team selected twenty sentinel sites; the initial sample frame was at the district

level from which the twenty sentinel sites were chosen. In order to oversample poor areas, the 5% of richest districts were excluded. Poverty level was determined by the Peruvian National Fund for Development and Social Compensation. The sampling strategy included rural and urban settings. To choose the sentinel sites, a multi-stage, cluster-stratified, random sampling technique was applied. Afterwards, one census tract in each district was randomly selected, and all block of houses and clusters of houses were counted. Finally, households in each selected block of houses or cluster of houses were searched to identify those with at least one child aged 6–18 months old, until a total of one hundred households were found. Exact details about the sampling procedures are published elsewhere (*Young Lives, 2008*).

## Statistical analysis

Analyses were conducted with STATA 11.0 (StataCorp, College Station, Texas, USA). Descriptive analyses were conducted using Chi-squared test to contrast categorical variables. Proportions and 95% confidence intervals (95% CI) were calculated. Means and standard deviations are presented as well. Cumulative incidence per 100 children-years and 95% CI were calculated for developing either of the outcomes of interest, after excluding those subjects who met the criteria for overweight or obesity at baseline. However, when assessing the cumulative incidence of central obesity no subject was excluded because of lack of data on waist circumference at baseline. Relative risk (RR) and 95% CI were calculated with generalized linear models assuming Poisson distribution, log link, and using robust standard errors to account for the cluster effect.

Regression analyses were conducted with participants with complete data in all variables included in the regressions; number of observations for each outcome is presented in Tables 2 and 3. Four models were constructed to assess the risk of interest using a hierarchical approach (*Victora et al., 1997*). We fit these models to understand which variables, child-related or family-related, alter or not the risk estimates at two points in time, early- and late-childhood. First, a crude model including only the outcome and exposure was fitted. Model A included adjustment by child-related variables: gender, age at baseline, breastfeeding, hospitalization after birth, birth weight and child birth order. Model B included adjustment by family-relates variables: household location and wealth index as well as maternal education and nutritional status; all the measures assessed at baseline (family-related variables). Model C included adjustment by all the previous variables. When the outcome was central obesity, the same four previous models were also fitted, though child nutritional status at baseline was included as a potential confounder in both models A and C.

The study used data of a prospective cohort study. Such longitudinal design could have allowed conducting more complex analysis including variables which values could have changed over time. We did not include such analysis because only a few of the variables could change over time, and of these only two were deemed to be important for the present study: wealth index and maternal BMI. Wealth index did not vary much between rounds (data not shown). Even though the frequency of maternal obesity increased over

time, when we included the current maternal BMI (e.g., maternal BMI at first follow-up when the outcome was at first follow-up too) the risk estimated did not change much in comparison to a model in which baseline maternal BMI was included (data not shown).

### Ethics

The Young Lives Study had ethical as a whole. Further details on the ethics procedures have been published elsewhere and are available online (*Young Lives, 2009*; *Young Lives, 2015*). Because this is a secondary analysis of a publicly available dataset, we did not request approval from an Institutional Review Board. Data can be accessed online: http://www.younglives.org.uk/ (*Barnett et al., 2013*). The analyses conducted in this study did not include personal information of any participant and only de-identified data was used.

## RESULTS

### Participants

There were 1,503 potential participants and 529 were excluded because of missing values: 974 children were included at baseline (Fig. S1). Comparisons between potential participants included and those excluded from the analyses (missing data) revealed these groups were not different in terms of gender, age, breastfeeding and delivery mode; there were however, differences between groups in family-related variables: maternal BMI and educational level, household wealth index and location (Table S1).

### Study population at baseline

Mean age at baseline was 11.7 ($\pm$3.5) months; and 50.1% were males; this population was predominately urban: 79.6%. Children born by Caesarean section were 15.6% (95% CI [13.3–17.9]). At baseline, the prevalence of overweight was 10.5% (95% CI [8.5–12.4]), whereas obesity was present in 2.4% (95% CI [1.4–3.4]). In the bivariate analyses Caesarean section was associated with nutritional status at baseline ($p = 0.008$). The distribution of other variables at baseline according to children nutritional status at baseline is shown in Table S2.

### Caesarean section and incidence of excess of weight

The number of children-years assessed at each time according to each outcome is presented in Fig. S2. Cumulative incidence of overweight, obesity and overnutrition at the first follow-up was: 3.2 (95% CI [2.8–3.7]), 1.1 (95% CI [0.9–1.4]) and 4.1 (95% CI [3.6–4.6]) per 100 children-years, respectively. Moreover, the cumulative incidence for these outcomes at the second follow-up was: 2.0 (95% CI [1.7–2.3]), 0.7 (95% CI [0.5–0.8]) and 2.5 (95% CI [2.2–2.9]), respectively. The cumulative incidence of abdominal obesity was 5.3 (95% CI [4.8–5.7]) at the second follow-up. The cumulative incidence for either outcome is detailed in Table 1.

The incidence of obesity was higher among children born by Caesarean section at first and second follow-up ($p < 0.001$). Overweight showed a similar pattern, with higher incidence for children born by Caesarean section at the first ($p = 0.006$) but not at the second ($p = 0.095$) follow-up.

**Table 1 Cumulative incidence at the first and second follow-up of overweight, obesity, overnutrition and central obesity. The Young Lives Study, Younger Cohort, Peru.**

| Variables | From baseline to first follow-up | | | From baseline to the second follow-up | | | |
|---|---|---|---|---|---|---|---|
| | Overweight | Obesity | Overnutrition | Overweight | Obesity | Overnutrition | Central obesity |
| | | | | Incidence (95% CI) | | | |
| *Mother* | | | | | | | |
| Maternal BMI | | | | | | | |
|   Normal weight | 2.7 (2.1–3.5) | 1.1 (0.7–1.6) | 3.5 (2.7–4.4) | 2.1 (1.7–2.7) | 0.7 (0.5–1.1) | 2.7 (2.2–3.3) | 5.3 (4.6–6.1) |
|   Overweight | 2.9 (2.1–4.0) | 1.5 (1.0–2.3) | 4.1 (3.1–5.5) | 2.9 (2.2–3.8) | 1.1 (0.7–1.6) | 3.8 (3.0–4.8) | 6.9 (5.8–8.1) |
|   Obesity | 4.3 (2.7–7.1) | 2.6 (1.4–4.7) | 6.5 (4.4–9.7) | 4.1 (2.7–6.2) | 2.11 (1.3–3.6) | 6.0 (4.3-8–.5) | 8.7 (6.8–11.2) |
| Maternal education | | | | | | | |
|   None/primary | 2.6 (1.8–3.7) | 0.3 (0.1–0.8) | 2.7 (2.0–3.8) | 1.9 (1.4–2.6) | 0.0 (0.0–0.3) | 1.9 (1.4–2.6) | 3.9 (3.2–4.8) |
|   High school | 2.9 (2.2–3.9) | 1.8 (1.2–2.5) | 4.3 (3.4–5.5) | 2.5 (1.9–3.2) | 1.4 (1.0–1.9) | 3.6 (2.9–4.5) | 7.0 (6.0–8.0) |
|   Higher education | 3.4 (2.3–5.0) | 2.1 (1.4–3.3) | 5.3 (3.9–7.2) | 4.0 (3.0–5.4) | 1.5 (1.0–2.3) | 5.2 (4.1–6.7) | 8.1 (6.7–9.8) |
| Wealth index | | | | | | | |
|   Bottom | 2.4 (1.5–3.8) | 0.6 (0.2–1.3) | 3.1 (2.1–4.5) | 1.5 (1.0–2.4) | 0.1 (0.0–0.5) | 1.4 (0.9–2.2) | 4.1 (3.2–5.3) |
|   Middle | 3.4 (2.5–4.6) | 0.6 (0.3–1.2) | 3.7 (2.8–5.0) | 2.1 (1.5–2.8) | 0.7 (0.4–1.1) | 3.0 (2.3–3.9) | 5.6 (4.7–6.7) |
|   Top | 2.8 (2.1–3.7) | 2.3 (1.7–3.1) | 4.7 (3.7–6.0) | 3.5 (2.8–4.3) | 1.7 (1.3–2.3) | 4.9 (4.0–5.9) | 7.8 (6.9–9.0) |
| Location | | | | | | | |
|   Urban | 2.9 (2.3–3.6) | 1.5 (1.2–2.0) | 4.1 (3.4–4.9) | 2.9 (2.4–3.4) | 1.2 (0.9–1.5) | 3.8 (3.3–4.5) | 6.9 (6.2–7.6) |
|   Rural | 3.0 (2.0–4.6) | 0.5 (0.2–1.4) | 3.6 (2.4–5.2) | 1.5 (1.0–2.4) | 0.2 (0.1–0.7) | 1.7 (1.1–2.6) | 3.6 (2.7–4.8) |
| *Children* | | | | | | | |
| Gender | | | | | | | |
|   Male | 3.2 (2.5–4.2) | 1.6 (1.1–2.2) | 4.6 (3.7–5.7) | 2.6 (2.0–3.2) | 1.1 (0.8–1.6) | 3.6 (2.9–4.3) | 4.1 (3.4–4.8) |
|   Female | 2.6 (1.9–3.5) | 1.1 (0.7–1.7) | 3.3 (2.6–4.3) | 2.5 (2.0–3.2) | 0.8 (0.5–1.2) | 3.1 (2.5–3.9) | 8.4 (7.4–9.4) |
| Age | | | | | | | |
|   <1 year | 3.1 (2.4–4.1) | 1.2 (0.8–1.9) | 4.0 (3.2–5.1) | 2.9 (2.3–3.7) | 1.1 (0.7–1.5) | 3.7 (3.0–4.5) | 6.5 (5.7–7.5) |
|   ≥1 year | 2.7 (2.1–3.6) | 1.4 (1.0–2.1) | 3.9 (3.1–4.9) | 2.3 (1.8–2.9) | 0.9 (0.6–1.3) | 3.1 (2.5–3.7) | 5.9 (5.1–6.8) |
| Birth weight | | | | | | | |
|   2,500–4,000 | 2.7 (2.2–3.4) | 1.3 (0.9–1.7) | 3.7 (3.1–4.5) | 2.6 (2.2–3.1) | 0.9 (0.7–1.2) | 3.3 (2.9–3.9) | 6.3 (5.6–7.0) |
|   ≤2,500 | 4.6 (1.7–12.3) | 0.0 (0.0–0.0) | 4.6 (1.7–12.3) | 2.9 (1.1–7.7) | 0.7 (0.1–5.1) | 3.6 (1.5–8.7) | 4.3 (1.9–9.6) |
|   ≥4,000 | 5.4 (3.1–9.6) | 3.3 (1.6–6.6) | 7.7 (4.8–12.4) | 2.9 (1.5–5.3) | 2.4 (1.2–4.5) | 4.9 (3.0–7.9) | 7.4 (5.2–10.6) |
| Breastfeeding | | | | | | | |
|   Yes | 2.9 (2.4–3.5) | 1.3 (1.0–1.8) | 3.9 (3.3–4.6) | 2.5 (2.2–3.0) | 1.0 (0.7–1.2) | 3.3 (2.9–3.8) | 6.2 (5.6–6.8) |
|   No | 3.2 (0.5–23.0) | 3.2 (0.5–23.0) | 6.5 (1.6–25.9) | 4.1 (1.0–16.5) | 2.1 (0.3–14.7) | 6.2 (2.0–19.2) | 8.3 (3.1–22.0) |
| Hospitalised after birth | | | | | | | |
|   Yes | 3.8 (2.2–6.5) | 1.6 (0.7–3.5) | 4.7 (2.9–7.6) | 3.4 (2.2–5.4) | 1.9 (1.1–3.4) | 4.7 (3.2–7.0) | 6.9 (5.1–9.4) |
|   No | 2.9 (2.3–3.4) | 1.3 (1.0–1.8) | 3.9 (3.3–4.6) | 2.5 (2.1–2.9) | 0.9 (0.7–1.2) | 3.2 (2.7–3.7) | 6.1 (5.5–6.8) |
| Birth order | | | | | | | |
|   1st | 3.1 (2.3–4.2) | 1.9 (1.4–2.7) | 4.5 (3.5–5.7) | 2.8 (2.2–3.6) | 1.2 (0.8–1.7) | 3.8 (3.0–4.7) | 6.8 (5.9–7.9) |
|   2nd | 1.9 (1.2–3.1) | 1.5 (0.9–2.4) | 3.3 (2.4–4.8) | 2.9 (2.1–3.9) | 1.2 (0.8–1.9) | 3.9 (3.0–5.0) | 6.8 (5.7–8.2) |
|   ≥3rd | 3.2 (2.3–4.4) | 0.5 (0.2–1.1) | 3.5 (2.6–4.8) | 2.0 (1.4–2.8) | 0.5 (0.2–0.9) | 2.4 (1.8–3.2) | 4.6 (3.8–5.7) |
| Delivery | | | | | | | |
|   No caesarean | 2.7 (2.1–3.3) | 1.0 (0.7–1.4) | 3.5 (2.9–4.2) | 2.5 (2.0–2.9) | 0.8 (0.6–1.0) | 3.1 (2.6–3.6) | 6.0 (5.4–6.7) |
|   Caesarean section | 4.4 (2.9–6.6) | 3.2 (2.1–5.0) | 6.7 (4.8–9.3) | 3.2 (2.2–4.7) | 2.2 (1.4–3.3) | 4.9 (3.6–6.7) | 7.2 (5.7–9.0) |

**Table 2 Risk of overweight, obesity and overnutrition provided the child was born by Caesarean section. The Young Lives Study, Younger Cohort, Peru.** Crude model, only include the outcome and exposure; Model A adjusted by child sex, child age at baseline, breastfeeding, hospitalised after birth, birth weight and child birth order; Model B adjusted only by location at baseline, maternal BMI at baseline, wealth index at baseline, and maternal education; Model C adjusted by all the previous variables. In bold, $p < 0.05$. C-sect: Caesarean section.

| | | RR (95% CI) | | |
|---|---|---|---|---|
| | Crude | Model A | Model B | Model C |
| | **From baseline to the first follow-up** | | | |
| | **Outcome: overweight** ($n = 644$) | | | |
| No C-sect | 1 | 1 | 1 | 1 |
| C-sect | **1.80 (1.14–2.84)** | **1.54 (1.00–2.36)** | **1.78 (1.09–2.89)** | 1.51 (0.98–2.35) |
| | **Outcome: obesity** ($n = 654$) | | | |
| No C-sect | 1 | 1 | 1 | 1 |
| C-sect | **3.12 (1.87–5.21)** | **2.59 (1.36–4.93)** | **2.46 (1.62–3.74)** | **2.25 (1.36–3.74)** |
| | **Outcome: overnutrition** ($n = 680$) | | | |
| No C-sect | 1 | 1 | 1 | 1 |
| C-sect | **1.90 (1.47–2.46)** | **1.63 (1.27–2.09)** | **1.81 (1.37–2.38)** | **1.57 (1.22–2.04)** |
| | **From baseline to the second follow-up** | | | |
| | **Outcome: overweight** ($n = 630$) | | | |
| No C-sect | 1 | 1 | 1 | 1 |
| C-sect | 1.35 (0.96–1.90) | 1.17 (0.83–1.64) | 1.29 (0.93–1.77) | 1.15 (0.84–1.55) |
| | **Outcome: obesity** ($n = 601$) | | | |
| No C-sect | 1 | 1 | 1 | 1 |
| C-sect | **2.69 (1.71–4.23)** | **1.96 (1.20–3.22)** | **2.02 (1.38–2.97)** | **1.57 (1.02–2.41)** |
| | **Outcome: overnutrition** ($n = 671$) | | | |
| No C-sect | 1 | 1 | 1 | 1 |
| C-sect | **1.48 (1.08–2.02)** | 1.25 (0.92–1.69) | **1.34 (1.02–1.75)** | 1.17 (0.92–1.50) |

**Table 3 Risk of central obesity provided the child was born by caesarean section. The Young Lives Study, Peru.** Crude model, only include the outcome and exposure; Model A adjusted by child gender, child age at baseline, breastfeeding, hospitalised after birth, birth weight, child birth order and child nutritional status at baseline; Model B adjusted only by location at baseline, maternal BMI at baseline, wealth index at baseline, and maternal education; Model C adjusted by all the previous variables. C-sect: Caesarean section.

| Variable | | RR (95% CI) | | |
|---|---|---|---|---|
| | Crude | Model A | Model B | Model C |
| | **From baseline to the second follow-up** | | | |
| | **Central obesity** ($n = 801$)[a] | | | |
| No C-sect | 1 | 1 | 1 | 1 |
| C-sect | **1.21 (1.00–1.48)** | 1.12 (0.96–1.31) | 1.09 (0.94–1.27) | 1.06 (0.91–1.22) |

**Notes.**
[a] Did not exclude any overweight or obese children at baseline; no data on waist circumference.

### Caesarean section as a risk factor for excess of weight

Children born by caesarean section in comparison to children who were not, did not have higher risk of overweight; in addition, the risk of overnutrition was only higher at the first follow-up (RR = 1.57, 95% CI [1.22–2.04]). Relative to children who were not born by Caesarean section, those who were born by Caesarean section had higher risk of obesity at the first (RR = 2.25, 95% CI [1.36–3.74]) and second (RR = 1.57, 95% CI [1.02–2.41]) follow-up (Table 2). There was no risk of central obesity at the second follow-up (Table 3).

In the hierarchical approach, at the first follow-up, family-related variables had a stronger effect in diminishing the risk of obesity in comparison to children-related variables (Table 2). At the second follow-up, there was the opposite trend (Table 2).

## DISCUSSION

### Main findings

In a sample of Peruvian children drawn from and currently living in resource-limited settings, our results suggest a higher probability of developing obesity among those born by Caesarean section, compared to those who were not. The risk of having obesity among those born by a Caesarean delivery was over twofold in early-childhood, and 57% higher in late-childhood. This observation suggests that risk magnitudes wane over time, and yet caution is warned as confidence intervals for risk estimates at both follow-up periods overlapped. Relative to child-related variables, family-related variables decreased more the risk of obesity at the first, than at the second follow-up. This suggests that family-related factors, more than child-related ones, appears as potential avenues to intervene to prevent the increased risk of obesity at early-childhood among children born by Caesarean section.

### Previous studies

Caesarean section delivery appears to be a risk factor for turning obese (*Darmasseelane et al., 2014*; *Keag, Stock & Norman, 2014*; *Kuhle, Tong & Woolcott, 2015*) and evidence from observational studies suggest such risk is stronger at early ages (*Kuhle, Tong & Woolcott, 2015*; *Pei et al., 2014*). In the USA, *Rooney, Mathiason & Schauberger (2011)* reported over 200% risk of obesity; the outcome was measured when the children were in average 5 years old (*Rooney, Mathiason & Schauberger, 2011*). Other study in the USA showed similar results and the outcome was measured at 3 years of age (*Huh et al., 2012*). Just recently, *Mueller et al. (2015)* reported a cohort study conducted in New York with women who classified themselves as African-American or Dominican, and reported higher risk of obesity when the offspring was 7 years old. In Germany, they found higher odds of obesity at age 2 years yet not at age 6 or 10 years, providing evidence that the risk diminishes over time (*Pei et al., 2014*). In contrast, *Goldani et al. (2013)* in Brazil found there were higher odds of obesity at age 10–11 years, yet not at age 7–9 years (*Goldani et al., 2013*); it is worth mentioning that Brazilian settings from which these results were retrieved were very dissimilar: one was less rich than the other (*Goldani et al., 2013*). Not all previous studies have found higher risk of obesity. *Flemming et al. (2013)* in Canada did not report a significant association; nutritional status was assessed at age 10–11. *Ajslev et al. (2011)*

in Demark had similar results with BMI assessed at age 7 years in average. Another study in Brazil did not find higher prevalence of obesity; the outcome was assessed at age 4 years (*Barros et al., 2012*).

With regard to central obesity, a cross-sectional study in Iran reported similar results: no significant estimates after adjusting for potential confounders (*Salehi-Abargouei et al., 2014*). They included a smaller sample size ($n = 635$) enrolled from elementary schools (*Salehi-Abargouei et al., 2014*). Although it seems that Caesarean section delivery is not a risk factor for childhood central obesity, more studies should be conducted so to clearly prove this observation, especially as it seems to be biologically plausible (*Sanz & Moya-Perez, 2014*).

The different rates we report regarding childhood obesity and Caesarean section, in comparison to other studies, could be because our sample excluded the highest socioeconomic level, and most of Caesarean section (56.2%) were reported at the top wealth index tertile (data not shown). Overall, the risk of obesity among those born by Caesarean section delivery could be stronger in the Peruvian general children population. This warrants further studies with a general-population-based sample. Socioeconomic status, as well as obesity and Caesarean section rates, may influence the risk estimates.

Provided the gut microbiota is in the causal path from Caesarean section to childhood obesity (*Cani & Delzenne, 2009*; *Cani et al., 2008*; *Gronlund et al., 1999*; *Isolauri, 2012*; *Mackie, Sghir & Gaskins, 1999*; *Penders et al., 2006*; *Reinhardt, Reigstad & Backhed, 2009*; *Snedeker & Hay, 2012*), variables which directly affect the gut microbiota may play a role too: diarrhea of different etiologies (*Albert et al., 1978*; *Gorkiewicz et al., 2013*; *Hopkins, Sharp & Macfarlane, 2001*; *Hopkins, Sharp & Macfarlane, 2002*; *Monira et al., 2013*), and re-colonization with regular microbiota after diarrhoea (*Monira et al., 2012*). Diet patterns which determine shifts in the gut microbiota between subjects in different continents (*De Filippo et al., 2010*), and within the same country (*Abdallah Ismail et al., 2011*), could also explain the different results.

## Caesarean section and risk of obesity: a hierarchical approach

*Pei et al. (2014)* suggested that the effect of Caesarean section on childhood obesity might decrease as the child grows up and other factors, such as dietary habits or lack of physical activity, start playing a more important role in the onset of obesity (*Pei et al., 2014*). Our results support this hypothesis because the risk estimates were stronger at younger ages. Moreover, family-related variables diminished the risk of obesity at the first follow-up, more than it did at the second follow-up. This means that family-related variables play a greater role in the risk estimates at younger ages. Efforts should be taken to address family-related variables of young children born by Caesarean section to prevent early-childhood obesity.

## Strengths and limitations

The main strength of this study is the analysis of prospective data. It also benefits from a large sample size of children in resource-limited settings in a developing country. The study population has characteristics that have not been included in previous studies.

In addition, we included several confounders and central obesity as outcome. The multi-stage, cluster-stratified, random sampling technique is also a strength.

However, limitations must be highlighted too. First, we could not control for what has been reported to be an important confounder (*Flemming et al., 2013*): pre-pregnancy BMI or pregnancy weight gain; this limitation is shared with other published studies (*Goldani et al., 2011*; *Goldani et al., 2013*; *Rooney, Mathiason & Schauberger, 2011*; *Zhou et al., 2011*). If we had adjusted for any of these variables, the risk estimates could have been smaller or perhaps they would have not been statistically significant, as other authors have reported (*Ajslev et al., 2011*; *Barros et al., 2012*; *Flemming et al., 2013*). Second, some studies have accounted for maternal smoking status during pregnancy (*Ajslev et al., 2011*; *Barros et al., 2012*; *Flemming et al., 2013*; *Pei et al., 2014*), though we did not. However, this limitation should have not affected much our results as the prevalence of smoking among women is low in some Peruvian cities (*Champagne et al., 2010*; *Heitzinger et al., 2014*; *Medina-Lezama et al., 2008*); this reduces the probability of women to be active smoker during pregnancy. Third, we could not exclude children with central obesity at baseline. However, this could have a minor impact as it is very unlikely for a child younger than one year old to be centrally obese. A study in Mexico reported that the prevalence of abdominal obesity was 4.4% among children aged 2.7 years in average (*Salinas-Martinez et al., 2012*). Provided Mexico has higher rates of childhood obesity than Peru, we believe the prevalence of abdominal obesity among Peruvian infants or children at very early childhood should be rather low. To the best of our knowledge there are no studies which have reported central obesity among infants. Moreover, in order to overcome this potential limitation, when the outcome was central obesity we adjusted for children nutritional status at baseline. We decided for this procedure because preliminary analyses revealed these were the most conservative results: when we did not adjust for children nutritional status at baseline the results were very similar, and when we excluded obese children as per BMI at baseline the estimates were higher. Fourth, the source of information about delivery mode could be a limitation as it relied on a questionnaire applied to the mother. As this information was collected when the children were 6–18 months old, we believe there should have been low recall bias. Fifth, we do not consider having two different criteria for childhood obesity is a limitation. The criterion (IOTF) we used to define obesity at either follow-up is more conservative in comparison to other criteria (*Gonzalez-Casanova et al., 2013*; *Padula & Salceda, 2008*; *Ramirez et al., 2006*). Sixth, there could be selection bias: there were more obese mothers in the study group. If we would have not applied the exclusion criteria (prematurity), there would still be more obese mothers included in the analysis; this difference might have influenced delivery mode. However, we cannot verify if mothers were obese at birth; data on maternal weight was assessed between six and eighteen months after child birth. Lastly, there are medical and obstetric issues that may have contributed to a Caesarean section including, for example, gestational diabetes. To the best of our knowledge there are not nationally representative data on gestational diabetes. However, one study in a hospital in Lima reported that gestational diabetes was

not associated with higher odds of Cesarean section (*Ilave & Gutarra, 2009*). There was also no information if the Caesarean section was planned or an emergency procedure.

Further sources of bias may affect cohort studies (*Delgado-Rodriguez & Llorca, 2004*; *Yu & Tse, 2012*) and these must be addressed too. First, attrition bias (*Barnett et al., 2013*) may be ruled out as losses to follow-up at the first and second follow-up were 4.3% and 1.1%, respectively. Second, missing values in multivariable analysis could have added bias; however, post-hoc analysis not restricting the analysis to subjects with complete information revealed almost identical results (data not shown). Also, it is most likely that missing values follow a random pattern, so the effect of the bias would change the estimates towards the null value. Third, non-response bias was addressed when comparing included and excluded individuals. Fourth, non-differential misclassification bias should not raise concerns; especially as it has been suggested that when the exposure is binary the bias will move the estimates towards the null risk value (*Copeland et al., 1977*). Lastly, common causes of misclassification bias were ruled out due to the study design: (i) observer/interviewer bias, data collection followed a standardized procedure and variables of interest for the present study were unknown by the interviewers; (ii) low risk of reporting bias as the hypothesis tested in the present study was unknown by the participants at time of data collection.

## CONCLUSIONS

Our results suggest a higher probability of developing obesity, but not overweight, among children born by Caesarean section delivery, and such risk appears to be stronger at younger ages. Provided family-related variables play an important role in such risk at early-childhood, a possible venue for childhood obesity prevention could be to focus on families with a young child who has been born by Caesarean section delivery.

## ACKNOWLEDGEMENTS

The data used in this publication come from Young Lives, a 15-year study of the changing nature of childhood poverty in Ethiopia, India (Andhra Pradesh), Peru and Vietnam (www.younglives.org.uk). Young Lives is funded by UK aid from the Department for International Development (DFID), with co-funding from 2010 to 2014 by the Netherlands Ministry of Foreign Affairs, and from 2014 to 2015 by Irish Aid. The views expressed here are those of the author(s). They are not necessarily those of Young Lives, the University of Oxford, DFID or other funders.

The authors want to express their gratitude to Lorena Saavedra for her assistance on the early conception of the idea, to Chris Meinzen and to Christian Loret de Mola for their comments on early versions of the manuscript.

### Funding

RMC-L, JJM, AB-O, and the CRONICAS Center of Excellence in Chronic Diseases were supported by the National Heart, Lung, and Blood Institute Global Health Initiative

under the contract Global Health Activities in Developing Countries to Combat Non-Communicable Chronic Diseases (Project Number 268200900033C-1-0-1). AB-O is currently supported by a Wellcome Trust Research Training Fellowship in Public Health and Tropical Medicine (Grant 103994/Z/14/Z). The funders had no role in study design, data collection and analysis, decision to publish, or preparation of the manuscript.

## Grant Disclosures

The following grant information was disclosed by the authors:
National Heart, Lung, and Blood Institute Global Health Initiative: 268200900033C-1-0-1.
Wellcome Trust Research Training Fellowship in Public Health and Tropical Medicine: 103994/Z/14/Z.

## Competing Interests

Jaime Miranda is an Academic Editor for PeerJ.

## Author Contributions

- Rodrigo M. Carrillo-Larco and Antonio Bernabé-Ortiz conceived and designed the experiments, performed the experiments, analyzed the data, wrote the paper, prepared figures and/or tables, reviewed drafts of the paper.
- J. Jaime Miranda analyzed the data, wrote the paper, prepared figures and/or tables, reviewed drafts of the paper.

## Data Deposition

The following information was supplied regarding the deposition of related data:
http://www.younglives.org.uk.

## Supplemental Information

Supplemental information for this article can be found online at http://dx.doi.org/10.7717/peerj.1046#supplemental-information.

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
