# Peer review of "Delivery by caesarean section and risk of childhood obesity: analysis of a Peruvian prospective cohort"

_PeerJ, doi:10.7717/peerj.1046_

## Round 0.1 · original submission · Major Revisions

The following major modifications (besides other minor aspects raised by the reviewers) should be included in the manuscript in order to make it acceptable for publication:

- Some aspects of the experimental design should be clarified (see Reviewer 1 and Reviewer 2).
- Some recent publication on the topic should be considered (see Reviewer 2).
- Discussion should be improved (see Reviewer 1 and Reviewer 2).

·

Basic reporting

This is an interesting analysis on the Young Lives Study. However a more measured conclusion is required (line 233-234). Can this paper support this statement? Considerations must be given to medical & obstetric issues which may have contributed to C/S, additionally is it possible to state that mode of delivery influences obesity in late childhood? ( line 235.) If this is the believe of the authors then a rationale is required to explain this association. Overall a more measured discussion is required with adequate

Experimental design

Introduction; What was the rationale for looking at delivery mode and offspring obesity in later years. This needs to be fully explained to set the scene.
Please clarify data available regarding mode of delivery. Is it likely that some mothers had C/S due to medical issues for example gestational diabetes which would be associated with larger birth weight and potentially obesity in later life?. Also some mothers may have had planned C/S versus emergency. This needs to be fully considered by the authors. More detail is required regarding data available at birth

Validity of the findings

Line 108 discusses overnutrition as obesity plus overweight; what was the rational for this grouping as a secondary outcome. Is this the same as the primary outcome?
Clarification required on data available; was this data collected at the time of the study or was it recalled? (Line 128). Basic questions were asked to retrieve data; was this data retrieved from subjects or from an electronic medical record? THis is unclear and feel that this is required to add depth to the methods. More detail is required on methods of obtaining data and availability of data. Line 206 refers to nutritional status. clarification is required on how this was assessed/ Does this relate to obesity or is this referring to assessment of dietary intake at these time points. Dietary intakes is not mentioned which would be important as a cofounded to the development of obesity

Additional comments

Overall an interesting analysis, however a more measured conclusion and discussion is required

Reviewer 2 ·

Basic reporting

- There are two recent systematic reviews and meta-analyses (n=43 studies) that have investigated this association in children (Kuhle et al. 2015) and adults (Darmasseelane et al. 2014). Many of the statements that the authors make in the Introduction and Discussion (e.g. "it is not known whether the magnitude of risk actually diminishes over time" or "Prospective studies from different world regions are needed, as developing-country settings may have distribution of exposures that may influence the strength of association between Caesarean section and childhood obesity") should be discussed in light of the results from these two reviews rather than on a small number of selected studies.
- The English is acceptable but would benefit from a review by a native speaker
- The abbreviation BMI needs to be introduced
- Excluding children born preterm is in order but the justification in line 100 does not make sense
- The sentence in lines 120-121 does not make sense to me
- Line 134: "causal" not "casual"
- Lines 158-159: You can't call that incidence then, it's a prevalence
- Lines 174-177: This statement rather belongs in the Discussion or should be omitted at all. Also, the first part of the sentence (including other countries) has nothing to do with the argument that you are trying to make. Lastly, it should read "continuous variable".
- I think the term time-varying covariates is only used with survival analysis.
- Lines 187-191: The authors need to provide the appropriate references to demonstrate that the data are publically available.
- Throughout the text the abbreviation for confidence interval is reversed

Experimental design

- Why did the authors not use waist circumference cutoffs that are specific for the ages at follow-up (i.e. 5.5 years and 8 years)? Essentially the same cutoff is proposed for boys below and above 8 years.

Validity of the findings

- It is not clear to me why the authors chose to exclude children that were overweight or obese in infancy when they are interested in the association between C-section and childhood obesity. Excess weight in infancy is on the pathway to excess weight later in childhood. As it stands, the risk estimates are for the influence of C-section *after* infancy on children's weight, effectively removing any effect it may have had on the body weight in the first two years of life.
- If the attrition rate is so low (~5%) why is there such a high proportion of missing data (~25%)? There is a considerable difference in the C-section rate, maternal BMI, area of residence, and education between exlcluded and included mother-child pairs that the effect estimates in the sample will be biased and not be generalizable.

---

## Round 0.2 · accepted · Accept

I consider the autors have answered al the aspects raised by the reviewers and that the modfiications have improved the manuscript, making it acceptable for publication.